# Expression and Characterization of a Novel Cold-Adapted Chitosanase from Marine *Renibacterium* sp. Suitable for Chitooligosaccharides Preparation

**DOI:** 10.3390/md19110596

**Published:** 2021-10-21

**Authors:** Lin-Lin Zhang, Xiao-Hua Jiang, Xin-Feng Xiao, Wen-Xiu Zhang, Yi-Qian Shi, Zhi-Peng Wang, Hai-Xiang Zhou

**Affiliations:** 1College of Safety and Environmental Engineering, Shandong University of Science and Technology, Qingdao 266510, China; linlinzhsd@126.com (L.-L.Z.); xf.xiao@163.com (X.-F.X.); zhangwx19991@163.com (W.-X.Z.); Shiyiqian0715@163.com (Y.-Q.S.); 2Tobacco Research Institute, Chinese Academy of Agricultural Sciences, Qingdao 266101, China; xiaohuaaoe@163.com; 3Marine Science and Engineering College, Qingdao Agricultural University, Qingdao 266109, China

**Keywords:** chitosanase, cold adaptation, chitooligosaccharide, *Renibacterium* sp.

## Abstract

(1) Background: Chitooligosaccharides (COS) have numerous applications due to their excellent properties. Chitosan hydrolysis using chitosanases has been proposed as an advisable method for COS preparation. Although many chitosanases from various sources have been identified, the cold-adapted ones with high stability are still rather rare but required. (2) Methods: A novel chitosanase named CsnY from marine bacterium *Renibacterium* sp. Y82 was expressed in *Escherichia coli*, following sequence analysis. Then, the characterizations of recombinant CsnY purified through Ni–NTA affinity chromatography were conducted, including effects of pH and temperature, effects of metal ions and chemicals, and final product analysis. (3) Results: The GH46 family chitosanase CsnY possessed promising thermostability at broad temperature range (0–50 °C), and with optimal activity at 40 °C and pH 6.0, especially showing relatively high activity (over 80% of its maximum activity) at low temperatures (20–30 °C), which demonstrated the cold-adapted property. Common metal ions or chemicals had no obvious effect on CsnY except Mn^2+^ and Co^2+^. Finally, CsnY was determined to be an endo-type chitosanase generating chitodisaccharides and -trisaccharides as main products, whose total concentration reached 56.74 mM within 2 h against 2% (*w*/*v*) initial chitosan substrate. (4) Conclusions: The results suggest the cold-adapted CsnY with favorable stability has desirable potential for the industrial production of COS.

## 1. Introduction

Chitooligosaccharides (COS) are the hydrolytic products of the chitosan derived from the total or partial deacetylation of chitin which is the second-most ubiquitous polysaccharide in nature. COS have attracted much attention in recent years because of their versatile pharmacological activities and biological functions, thus applicability in various fields, such as antioxidant activity [1], anti-inflammatory activity [2], immunomodulation [3,4], cosmetic industry [5], agricultural industry [6,7], and so on. COS can be prepared by physical, chemical, and enzymatic degradation methods. However, chemical production of COS is environmentally hazardous and generally difficult to produce specific COS, always accompanied by the generation of mixed oligosaccharides with varying degrees of polymerization (DPs) [7]. Therefore, environmentally compatible and reproducible alternatives for preparation of COS are desirable. Enzymatic method is an excellent alternative to conventional processes, which can yield better-defined COS or degraded chitosan with desired physicochemical and biological properties [8].

The enzymes for the depolymerization of chitosan include the specific enzymes called chitosanases and some non-specific enzymes, such as carbohydrases and proteases [9]. As a type of glycoside hydrolase, chitosanase (EC 3.2.1.132) catalyzes the hydrolysis of β-1,4-linked glycosidic bond of chitosan and release COS as major product. According to the carbohydrate-active enzymes (CAZy) database, chitosanases can be grouped into several different glycoside hydrolyase (GH) families, among which GH46, 75, and 80 families contain chitosanases exclusively [10]. Most reported chitosanases belong to GH46; the members of this family have been characterized most extensively and have a highly electronegative substrate-binding cleft compared with other chitosanases [10]. Thus far, chitosanases have been found in many organisms including bacteria, fungi, plants, and viruses [11]. Most of these characterized chitosanases come from the terrestrial environment; only a few reports referred to the marine-derived chitosanases [11]. Marine microorganisms are endowed with unique genetic structures by the marine environment, and as a result are considered as new promising sources of the enzymes with unsuspected application potential [11]. For example, a GH46 chitosanase CsnA identified from *Renibacterium* sp. QD1 from the coast of Qingdao, China, displayed a broad pH stability of 5.0–10.0 [12]. An endo-type chitosanase CHIS5 mined from the metagenome of marine microorganisms has been used to degrade acetylated chitosan for efficient production of COS associating with the chitin deacetylase CDA20 [13].

To date, a variety of chitosanases have been identified; nevertheless the enzymes with special properties, such as cold-adaption, thermo-tolerance and single product distribution, are still few but required for industrial production [14]. In the past few years, some of the cold-adapted chitosanase-producing microorganisms have been isolated, such as *Bacillus* sp. BY01 and *Pseudoalteromonas* sp. SY39 [15,16]. Cold-adapted enzymes show high relative catalytic activities at low temperatures (generally less than 30 °C); even the optimal temperature may be higher [17]. Compared with the mesophilic enzymes, cold-adapted ones have evolved a range of structural features with high level of flexibility, which confer strong adaptation at low temperatures onto the enzymes [16,17]. These characteristics make running enzymatic processes possible at room or even lower temperatures without heating, which are conducive to reducing energy costs and contamination risks, and enhancing reaction process controllability in industrial production [18,19]. Therefore, cold-adapted enzymes have been found to be attractive in bioconversion.

Herein, a new cold-adapted chitosanase CsnY from the marine bacterium *Renibacterium* sp. Y82 was purified and characterized after heterologous expression by *Escherichia coli*. CsnY was identified as an endo-type chitosanase by hydrolytic products analysis and also had excellent thermo-tolerance properties. Noticeably, further biochemical characterization showed that almost all common metal ions or chelators had no obvious influence on the enzymatic activity of CsnY. These properties of CsnY suggest it could be regarded as excellent potential candidate for industrial applications of COS production.

## 2. Results

### 2.1. CsnY Sequence Analysis

In previous work, the marine bacterium *Renibacterium* sp. Y82 displayed the ability to degrade chitosan and grow in a chitosan sole-carbon medium (detailed data not shown). The genomic analysis of Y82 implied there existed a putative chitosanase-encoding gene *csnY* (Genbank number MT741946), the ORF of which consisted of 945 bp, encoding 314 amino acids with a signal peptide of 60 amino acids (Met^1^–Ala^60^) in the N-terminal (Figure 1). The amino acid sequence analysis based on the Conserved Domain Database (CDD) from the National Center of Biotechnology Information (NCBI, Bethesda, MD, USA) showed that CsnY had a conserved domain feature as a lysozyme-like superfamily of chitosanase-glyco-hydro-46 site, marked with a black line in Figure 1. Therefore, CsnY was determined as a member of the GH46 family, which is further classified into five different clusters from A to E. The phylogenic analysis result displayed CsnY was affiliated to Cluster A and originated from the same ancestral node with the chitosanase from *Renibacterium salmoninarum* (Genbank number WP_041684833.1) (Figure 2). The mature enzyme of CsnY protein had a calculated Mw of 27.8 kDa and pI value of 5.98. Multiple sequences alignment among CsnY and other typical chitosanase proteins of Cluster A indicated that CsnY shared 60.65% amino acid identity.

### 2.2. Expression and Purification of CsnY

The *csnY* nucleotide sequence was optimized without the signal sequence and shown in Figure 3. Heterologous expression of CsnY fused with 6×His-tag in the C-terminal was performed in *E. coli* BL21 (DE3). After 20 h of IPTG induction, the recombinant CsnY was purified via Ni–NTA affinity chromatography and finally the activity reached 369.31 U/mL after concentration through ultrafiltration, with a specific activity of 330.67 U/mg. The recombinant CsnY showed a single band located in SDS-PAGE gel with around 30 kDa (Figure 4), which was a little higher than the deduced Mw (27.8 kDa) of the mature enzyme, on account of fusing with 6×His-tag and addition of the restriction sites to the ends of the gene for construction of the plasmid, which introduced 10 extra amino acids into recombinant CsnY.

### 2.3. Effects of Temperature and pH on CsnY Activity and Stability

The pH stability of chitosanase CsnY was assessed by the measurement of residual enzymatic activity after incubation at various pH values for 24 h at 4 °C. CsnY kept stable at the range of pH 5.0–9.0, but lost its most of activity when the pH values were out of this range (Figure 5a). Figure 5b indicated that the optimum pH of CsnY was 6.0 in the phosphate buffer and the enzyme maintained relatively high activity between pH 5.0 and 8.0.

Thermal stability of CsnY was analyzed after incubation of enzyme for 0.5 h and 1 h under a variety of temperatures; the results illustrated CsnY possessed favorable thermostability below 50 °C and remained above 80% activity after incubation for 1 h (Figure 5c). As shown in Figure 5d, CsnY had an optimal temperature at 40 °C and contained over 80% activity in the range of 20–50 °C, especially at low temperatures (20–30 °C), demonstrating the cold-adapted property. Noticeably, about 60% activity was maintained even if the temperature was as low as 10 °C (Figure 5d). Furthermore, the thermal stability of CsnY within 12 h was also determined and the results were shown in Appendix A. Although CsnY displayed the optimum activity at 40 °C, it seemed the thermostability of which could not sustain for longer time. The half-life at this temperature was less than 3 h, and almost all the activity was lost after 6 h (Appendix A). However, the thermostabilities were quite better at 20 and 30 °C, the half-lives of CsnY at these temperatures were estimated to be over 6 and 4 h, respectively, and 30.2% and 21.3% relative activities were remained even after 12 h (Appendix A), indicating the industrial availability of CsnY at room temperature.

### 2.4. The Effects of Various Metal Ions or Chemicals on CsnY

In order to investigate the effects on CsnY activity of various metal ions and chemicals, the retained activities were monitored in the presence of these substances with the concentration of 1 mM. Figure 6 showed that the activity of CsnY had a 2.78-fold increase when Mn^2+^ was added into the reaction system. On the contrary, Co^2+^ had an obvious inhibitory influence on the activity of CsnY and resulted in over 35% loss of enzyme activity (Figure 6).

### 2.5. Final Degradation Product Analysis

Considering the effect on enzymatic reaction brought by high viscosity of polysaccharides, 2% (*w*/*v*) chitosan solution was used as the substrate. The depolymerization reaction was performed at room temperature (approximately 25 °C) by using cold-adapted CsnY, and the amount of reducing sugar in the reaction mixture was determined periodically. After hydrolysis for 1 h, the increase in reducing sugar became painfully slow, and finally, the concentration of which was determined as 56.74 mM at 2 h (Appendix A). Thin-layer chromatography (TLC) and positive-ion electrospray ionization mass spectrometry (ESI-MS) were applied to analyze the final hydrolysis products, and the results were displayed in Figure 7. At the end of the reaction, two clear spots corresponding to chitobiose and -triose, respectively, of markers were clearly observed on the TLC plate (Figure 7a). Therefore, the final depolymerization products of CsnY towards chitosan contained chitotrisaccharide and -disaccharide. As shown in Figure 7b, two main ion peaks with the mass-to-charge ratios (*m/z*) of 341.3 and 502.3 appeared under the positive mode, corresponding to [DP2 + H]^+^ and [DP3+H]^+^, respectively, and the peak of 363.3 *m/z* presumably belonged to [DP2 + Na]^+^. Therefore, disaccharinde and trisaccharide were proven to be the main products in accordance with the TLC analysis result. Interestingly, almost all of the chitosan substrates were degraded by CsnY, and no oligosaccharide with DP > 3 was detected by both TLC and ESI-MS methods (Figure 7). Moreover, the releasing mode of products and the rapid decrease in the viscosity of the mixture (data not shown) suggested that the chitosanase CsnY was active in an endolytic manner.

## 3. Discussion

Sequence analysis result revealed that several essential amino acid residues existed in CsnY and were marked by green star in Figure 2, among which two highly conserved residues (Glu^98^ and Asp^116^) were required for catalysis. Owing to the previous research, during the hydrolysis process, the Glu protonated the glycosidic oxygen and the Asp polarized the attacking water in the inverting catalytic reaction [20,21]. Experimentally confirmed by Lacombe-Harvey et al., Asp^116^, Arg^118^ and Thr^121^ residues were strictly conserved in chitosanases and essential for catalysis [22]. Additionally, other several residues as the members of ionic interaction network, including Asp^222^, Arg^267^ and Arg^282^, could stabilize the catalytic cleft with each other [21].

The biochemical properties of recombinant CsnY were surveyed. CsnY had a stable activity at the range of pH 5.0–9.0 (Figure 5a), which was similar with that of other bacterial chitosanases, such as Csn21c, CsnM, and CsnQ, displayed a stable activity when the pH values were from 4.0 to 9.0 [16,23,24]. The chitosanase isolated from *Renibacterium* sp. QD1 maintained stable at pH 5–10 where more than 90% of original activity was retained [12]. The optimum pH of the chitosanase CsnY was determined as 6.0 (Figure 5b). As previously reported, the optimal pH values for most of chitosanases were located at acidic or neutral range (pH 4–7) [23,25]. For example, the chitosanase Csna expressed the highest activity at the range of pH 5.3–6.0, and some other chitosanases, such as BaCsn46B and the chitosanase from *Paenibacillus mucilaginusus* TKU032, displayed the analogous optimum pH values as Csna [1,26].

The optimum temperature of cold-adapted CsnY was 40 °C (Figure 5d). The optimal temperatures of cold-adapted enzymes are generally below 35 °C [17,18,19]. Previous reports showed that the cold-adapted chitosanases GsCsn46A and Csn-CAP displayed the maximum activity at 30 °C [27,28]. However, some other chitosanases with cold-adapted property had the highest activity at the temperatures above 35 °C (Table 1).

Similar to CsnY, CsnM also showed the optimal activity at 40 °C [16]. Interestingly, CsnY possessed wider temperature range for the activity compared with other chitosanases (see Table 1 for details), with relatively high activity (over 80%) at the temperatures ranging from 20 to 50 °C, especially at low temperatures (20–30 °C) (Figure 5d). Noticeably, about 60% activity was maintained even if the temperature was as low as 10 °C (Figure 5d), which is rare among all the reported chitosanases. As the cold-adapted chitosanases, CsnS and CsnB retained 42.6% and 40.4% of its maximum activity, respectively, at 10 °C [15,29], GsCsn46A maintained 70% of its initial activity but with worse thermostability and lower specific activity compared with CsnY [28]. These results demonstrated that CsnY had an excellent cold-adapted property, which could be used at room temperature (generally 25 °C) or even lower temperatures to run biocatalytic processes without heating, in order to save energy and production costs, and to reduce contamination risks [16]. Although sharing the same optimal temperature with some other cold-adapted chitosanases (such as CsnM, CsnB, and the chitosanase from *Janthinobacterium* sp. 4239), CsnY exhibited much higher relative activity than those enzymes under 10 to 30 °C (Table 1), signifying better cold-adapted property owned by CsnY, which revealed a superior potential for application to prepare COS at room temperature. The chitosanases properties have a close relationship with the growth conditions of the microorganisms which produce chitosanases; the temperatures of the marine environments are generally relatively low, and thus the enzymes derived from marine microorganisms such as CsnY often possess particular properties such as cold adaptation, helping these microorganisms better adapt to outside environments [31]. Cold-adapted enzymes usually have a higher degree of flexibility, particularly in the region of active sites [19]. Their catalysis reactions invariably show smaller enthalpy and more negative entropy of activation, the reduction in the activation enthalpy weakens the temperature dependence of the reaction rate, which thus facilitates catalysis at low temperature [19]. Therefore, the chitosanase CsnY with the cold-adapted property may be relevant with protein flexibility, and perhaps another work related to the three-dimensional structure of CsnY will be carried out in the future to further elucidate why it possesses such excellent cold-adapted properties.

In addition, the CsnY exhibited higher thermostability than other cold-adapted chitosanases over a broad temperature range of 0 to 50 °C (see Table 1 for details). Due to catalysis reactions conducted by cold-adapted enzymes usually at low temperatures (20–30 °C), they are supposed to have lower thermostability than their mesophilic homologs, rapid deactivation occurs even when the environmental temperature increases slightly [15,18,19]. However, CsnY showed thermo-stable property and industrial availability within at least 12 h (Appendix A), which is beneficial for extended use, storage, and transport of the enzyme.

The activity of CsnY could be significantly enhanced by Mn^2+^ but inhibited by Co^2+^ (Figure 6). For some chitosanases, such as CsnB, Csn-CAP, CSN, and CsnW2, Mn^2+^ has obviously enhanced their enzymatic activities, with an increase of 2.57, 1.89, 1.9, and 1.16-fold, respectively [15,27,32,33]. The increases in chitosanase enzymatic activities can be explained that Mn^2+^ favors the saccharification process resulting in higher production of reducing sugars, GH-46 family chitosanases have the metal ion binding sites where some metal ions such as Mn^2+^ may bind to these sites and thus, helpful to enhance the three-dimensional structural stabilities, and catalytic activities of chitosanases [15]. Similar with CsnY, Co^2+^ also had an inhibitory impact on CsnM [16]. Surprisingly, distinguished from other chitosanases, no obvious enzyme activity changes were observed when other kinds of metal ions or chemical reagents, including Fe^3+^, Cu^2+^, EDTA, and SDS, were added into the reaction systems (Figure 6). The results above illustrated that CsnY could still keep a high stability under the influence of different metal ions and chelators, and had a desirable potential for industrial application.

CsnY degraded 2% (*w*/*v*) chitosan efficiently and thoroughly within 2 h releasing chitodisaccharides and -trisaccharides as main products with the total concentration of 56.74 mM (Appendix A and Figure 7), almost all of the substrates were degraded by CsnY within 2 h, and no oligosaccharide with DP > 3 was found by both TLC and ESI-MS methods (Figure 7), demonstrating the strongly depolymerized ability of CsnY against chitosan, which benefited from favorable catalytic activity. The specific activity of CsnY was determined as 330.67 U/mg, although this activity was detected not under the optimal condition, which was relatively high among cold-adapted chitosanases (Table 1), and able to satisfy the application in the industrial preparation of COS. Obviously, more chitosan could be used during COS production by CsnY, but the influence on enzymatic hydrolysis brought by high viscosity of polysaccharides appeared when increasing the concentration of substrate. Fortunately, CsnY could at least retain its stability at room temperatures (20–30 °C) for a longer time (Appendix A). Therefore, maybe it would be better adding the chitosan substrate in batches. The systematic research of COS preparation by CsnY will be performed in another work. Similarly, previous research showed that a number of cold-adapted chitosanases, which were determined as endo-type enzymes, catalyzed the cleavage of β-1,4-glycosidic linkage and released chitobiose and -triose (Table 1). During the enzymatic production of COS, simple product distribution would be significant for the following separation and purification. Some chitosanases reported previously degraded substrate with mixed COS of DP2–5 as the final products [32,34,35], thus compounding the difficulty for separation. However, although many cold-adapted chitosanases shared the same final products to CsnY, the production efficiency of them were far less [15,16], especially for CsnM, which thoroughly degraded chitosan substrate until 24 h later at 30 °C [16]. The high degradation specificity of CsnY is propitious to high-efficiency production of COS, combined with the desirable cold adaptation, favorable stability, strong resistance to most ions and chemicals; all of these outstanding properties suggest CsnY would be a potent tool for industrial production.

## 4. Materials and Methods

### 4.1. Materials, Strains, Plasmids, and Media

Chitosan (viscosity: 200 mPa·s; deacetylation degree: 95%) was purchased from Aladdin Biochemical Technology Co., Ltd. (Shanghai, China). Standard chitosan trisaccharide, disaccharide, and monosaccharide were purchased from Qingdao BZ Oligo Biotech Co., Ltd. (Qingdao, China). The TLC silica gel plates 60 F254 were bought from Merck KGaA (Darmstadt, Germany).

The marine bacterium strain *Renibacterium* sp. Y82 was isolated from brown seaweed in the Yellow Sea, China. In brief, the chips of rotten brown seaweed were put into the chitosan sole-carbon medium. After culture in the flask for microorganism enrichment and spread plate cultivation for isolation, a *Renibacterium* sp. strain named Y82 was found to grow in the chitosan sole-carbon medium, signifying the ability to degrade and apply chitosan (detailed data not shown), and stored in the laboratory. *E. coli* DH5α and BL21 (DE3) were used for plasmid construction and *csnY* gene expression, respectively. Both these strains were cultured at 37 °C in Luria-Bertani (LB) broth or solid medium with 2% (*w*/*v*) agar, into which 50 μg/mL kanamycin was supplemented if necessary. Expression vector pET-28a (+) was purchased from Novagen (Madison, WI, USA).

### 4.2. Sequence Analysis of CsnY

Based on the genomic analysis of *Renibacterium* sp. Y82 (relevant data not published), a putative gene named *csnY* which encoding a chitosanase CsnY was found and deposited in the Genbank getting the accession number MT741946, 22 May 2021. The conserved domain and the signal peptide of CsnY were analyzed by the Conserved Domain Database (CDD) (https://www.ncbi.nlm.nih.gov/cdd) and SignalP 5.0 server (http://www.cbs.dtu.dk/services/SignalP/) [36], respectively. The theoretical pI and Mw of CsnY were calculated using the compute pI/Mw Tool (https://web.expasy.org/compute_pi/). To clarify the evolutionary relationship among CsnY and other chitosanases from bacteria, the phylogenetic tree construction was executed by the neighbor-joining method with MEGA 6.0 software, according to the protein sequences of related chitosanases obtained from National Center of Biotechnology Information (NCBI), Bethesda, MD, USA (https://www.ncbi.nlm.nih.gov/) [37]. The multiple sequences alignment was created by means of DNAMAN software (Lynnon Biosoft, Foster City, CA, USA) [38].

### 4.3. Codon Optimization and Construction of Expression Vector

Following elimination of the rare codons used in *E. coli* and addition of *Nco*I and *Xho*I restriction sites to the terminals (Figure 3), the *csnY* gene (762bp, see Figure 3 for detailed sequence) without signal sequence and stop codon was synthesized by Synbio Technologies LLC (Synbio Technologies, Suzhou, China). The synthesized DNA was digested through restriction endonucleases *Nco*I and *Xho*I, afterwards, the digested *csnY* gene fragment was purified and ligated into the corresponding sites of the plasmid pET-28a (+) using T4 DNA ligase to construct the final recombinant plasmid with a C-terminal 6×His-tag, which was then transformed into *E. coli* BL21 (DE3) cells for enzyme expression.

### 4.4. Expression and Purification of CsnY

The recombinant *E. coli* BL21 (DE3) harboring the *csnY* gene was cultured in the LB broth supplemented with 50 μg/mL kanamycin at 37 °C till OD_600nm_ reached around 0.8. CsnY induction expression was carried out by using 0.5 mM IPTG for 20 h at 16 °C and 180 rpm. Cells were collected via centrifugation and suspended in 50 mM phosphate buffer (pH 7.0), and then sonicated at 4 °C. The crude protein supernatant obtained through centrifugation was injected into the Ni-NTA agarose column (TaKaRa, Dalian, China) which was already equilibrated by 50 mM phosphate buffer (pH 7.0) containing 300 mM NaCl and 15 mM imidazole. The column was then successively subjected to the same buffer with 30 mM imidazole to wash the sample and remove protein impurities, and to the same buffer with a linear gradient of imidazole (50–400 mM) to elute the 6×His-tagged CsnY. The eluted fractions with chitosanase activity were collected and pooled, then concentrated with a Millipore centrifugal filter 3 K device (Millipore, Burlington, MA, USA), meanwhile, the enzyme solution was desalted and 50 mM Tris-HCl (pH 7.0) was employed to replace the phosphate buffer. 12% of SDS-PAGE system (Bio-Rad, Hercules, CA, USA) was applied to analyze the Mw of recombinant CsnY, the loading amount was set as 10 μL. The protein marker was bought from Solarbio Life Sciences (Beijing, China). BCA protein assay kit (Solarbio, Beijing, China) was used to measure the total protein concentration.

### 4.5. Measurement of CsnY Activity

The chitosanase activity was detected at 40 °C and pH 7.0 according to the description by Zhou et al. [16]. In brief, enzymatic reaction was performed with 100 μL properly diluted CsnY solution and 900 μL substrate solution of 0.3% (*w*/*v*) chitosan for 10 min. Then, the reducing sugars were determined using 3,5-dinitrosalicylic acid (DNS) method [39]. One unit of chitosanase was defined as the amount of enzyme that generated reducing sugars corresponding to 1 μmol of glucosamine hydrochloride per min.

### 4.6. Effects of Temperature and pH on CsnY Activity and Stability

A total amount of 20 mM of four different buffers, including NaAc-HAc (pH 3.5–5.5), phosphate buffer (pH 5.5–8.0), Tris-HCl (pH 7.0–8.5), and Gly-NaOH (8.5–10.0), were used to evaluated pH stability of CsnY by measuring the residual enzymatic activities after incubation at 4 °C for 24 h, here taking the highest residual activity as 100%. To determine the optimum pH of CsnY, the enzymatic activities were measured in the four different buffers (pH 3.5–10.0) mentioned above at 40 °C by taking the activity at optimal pH as 100%. Thermostability of CsnY was assessed over the range of 0–70 °C, residual enzyme activities were detected after 0.5 and 1 h of incubation and calculated with the initial activity as 100%. The optimal temperature of the purified enzyme was determined over the range of 0–80 °C by setting the activity at the optimum temperature as 100%.

In order to assess the ability of CsnY for catalytic hydrolysis in a longer time, the thermal stabilities of CsnY within 12 h was investigated on the basis of the residual activities of CsnY measured at various time intervals during incubation at certain temperatures (20 °C, 30 °C, and 40 °C).

### 4.7. The Effects of Various Metal Ions or Chemicals on CsnY Activity

Different kinds of metal salts and chemical reagents with the concentration of 1 mM were added into the standard reaction solution to investigate their effects on CsnY activity. The relative activities were calculated with respect to the control sample where the reaction was conducted in the absence of any additive.

### 4.8. Analysis of Degradation Products

For enzymatic product analysis, 2% (*w*/*v*) chitosan solution as the substrate, was mixed with excess CsnY (10 U per mg of chitosan). The mixture was continuously stirred at room temperature (approximately 25 °C) to guarantee the depolymerization reaction was performed. Reducing sugar content in the reaction mixture was detected periodically by DNS method [39], to certify the degradation products no longer changed. Then, the mixture solution was concentrated using a centrifugal filter 3 K device (Millipore, Burlington, MA, USA), meanwhile, the proteins and undegraded macromolecules were removed. The end product of CsnY against chitosan was analyzed by means of TLC, following the previously described method [16,40]. Briefly, the samples on the plate were developed with a mixture of ammonia, water, and isopropanol (3:27:70, *v*/*v*/*v*) as the developing solvent, and visualized after drying the plate by spraying with 0.5% (*w*/*v*) ninhydrin in ethanol, and heating at 80 °C for 20 min. To further investigate the DPs of the oligosaccharides in the final product, the product solution was mixed with methanol (1:1, *v*/*v*), then quantitatively injected into an ESI-MS instrument (Bruker Esquire HCT, Billerica, MA, USA). The hydrolytic products were profiled in positive-ion mode under the following conditions: calibration dynamics, 2; cone voltage, 20.00 V; capillary voltage, 4.00 kV; desolvation temperature, 350 °C; source temperature, 150 °C; desolvation gas flow, 500 L/h; cone gas flow rate, 50 L/h; scan range, 100–1500 *m/z* [38].

## 5. Conclusions

In this work, a novel GH46 family chitosanase CsnY from marine bacterium *Renibacterium* sp. Y82 was heterologously expressed, purified, and characterized. Significantly, CsnY was a cold-adapted enzyme with favorable stability, especially showing high relative activity at low temperature (10–30 °C). Moreover, most metal ions or chemicals had no obvious influence on the enzymatic activity of CsnY, which released chitodisaccharide and -trisaccharide from chitosan. These properties suggest CsnY would be an excellent potential candidate for industrial preparation of COS. The future works will focus on the three-dimensional structure analysis of CsnY to further elucidate the relationship between the structure and its excellent cold-adapted property, and the systematic study of COS preparation by CsnY.

## Figures and Tables

**Figure 1 marinedrugs-19-00596-f001:**
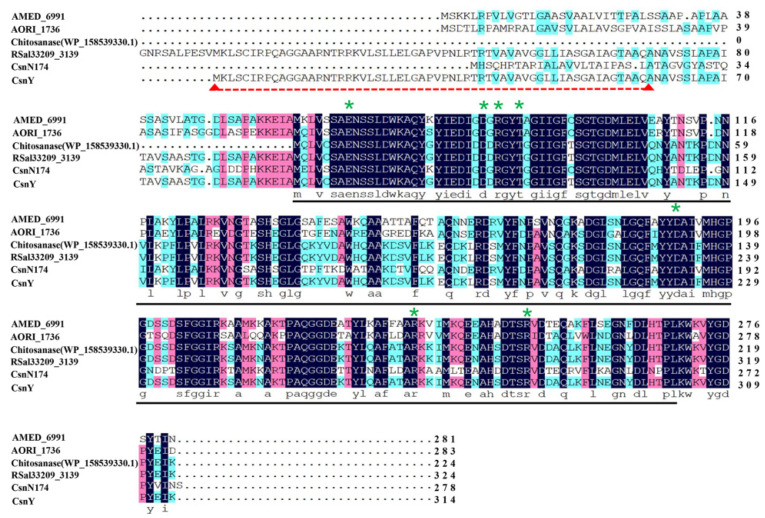
Multiple sequences alignment among CsnY and other related chitosanases of GH46 family. AMED_6991 (GenBank number: ADJ48710.1), AORI_1736 (GenBank number: AGM04324.1), chitosanase (GenBank number: WP_15853930.1), RSal33209_3139 (GenBank number: ABY24857.1), CsnN174 (GenBank number: AAA19865.1), CsnY (GenBank number: MT741946). The signal peptide was underlined with red dotted line. The key residues for catalysis and stabilization were labeled with a green star. The conserved sequence with the characteristic of lysozyme-like superfamily was marked with a black line.

**Figure 2 marinedrugs-19-00596-f002:**
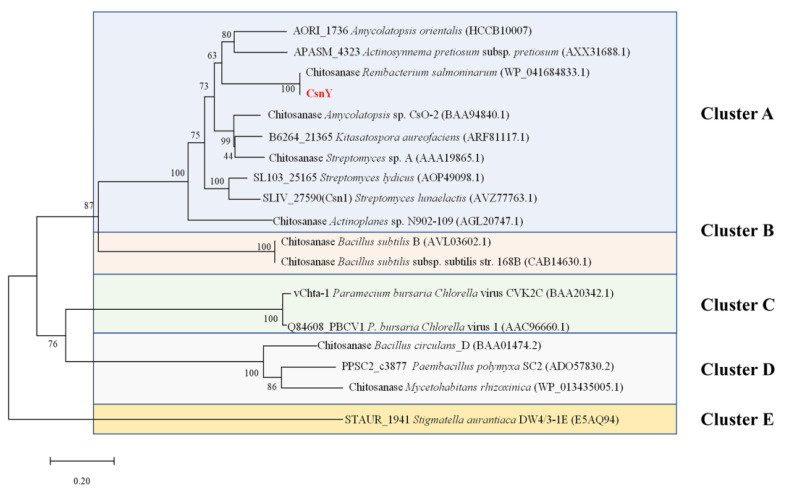
Phylogenetic analysis of CsnY and other chitosanases which belong to GH46.

**Figure 3 marinedrugs-19-00596-f003:**
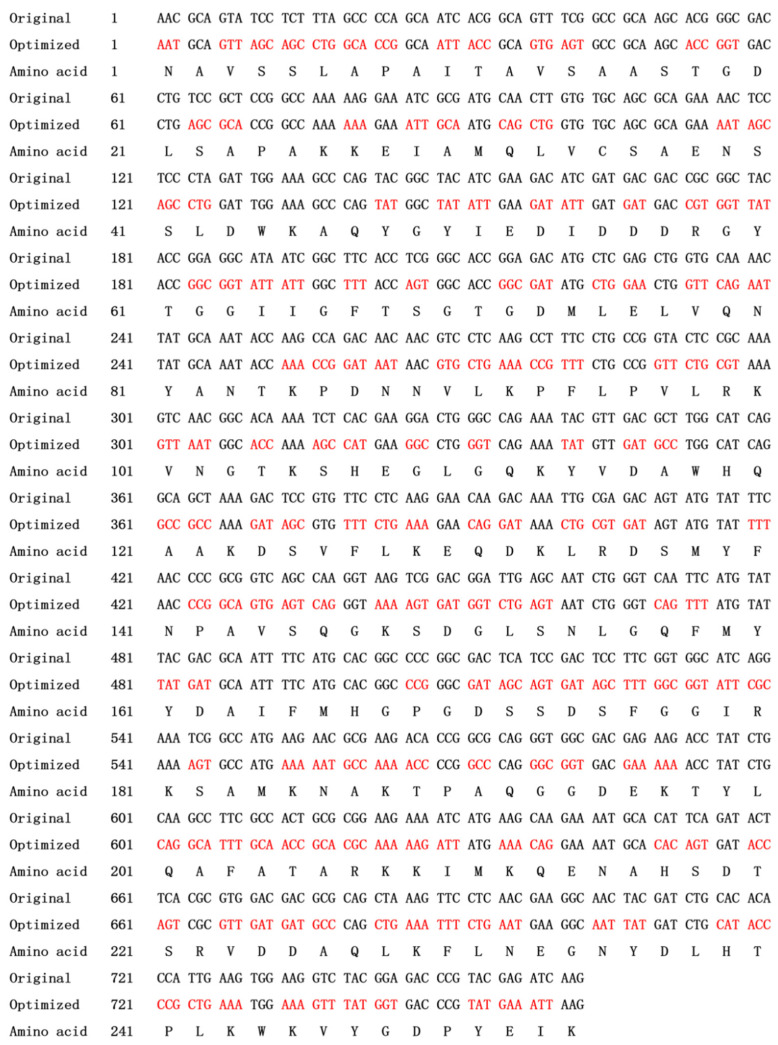
The original *csnY* sequence and the optimized sequence based on *E. coli* without signal sequence or stop codon. The mutated codons were highlighted in red; the deduced amino acid sequence was displayed below.

**Figure 4 marinedrugs-19-00596-f004:**
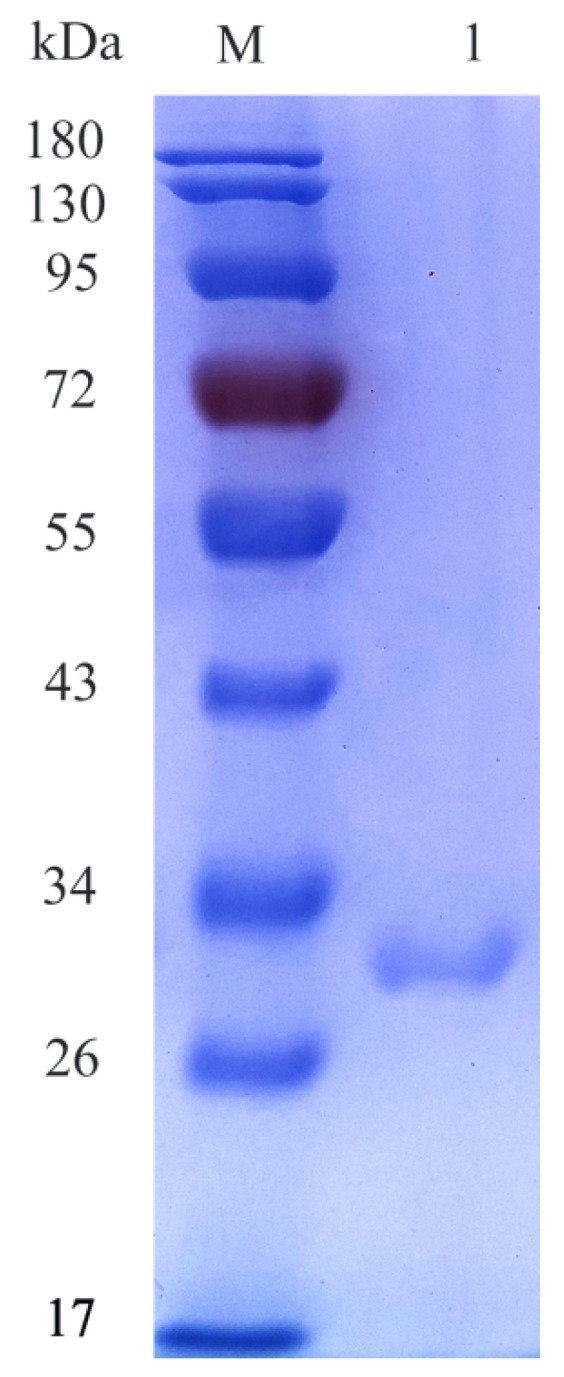
SDS-PAGE analysis of recombinant chitosanase CsnY. Lane M: standard Mw markers; lane 1: purified CsnY.

**Figure 5 marinedrugs-19-00596-f005:**
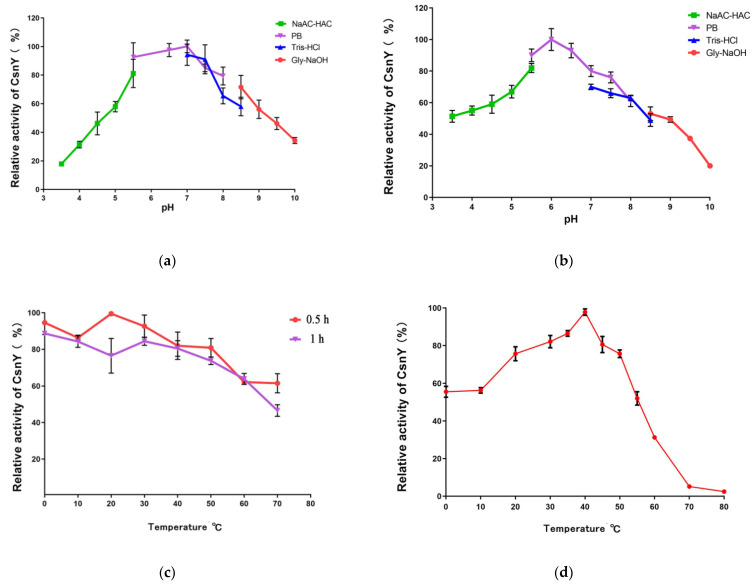
Effects of pH and temperature on the activity and stability of CsnY. (**a**) The pH stability of CsnY was analyzed after incubation for 24 h in the pH range of 3.5 to 10.0, with the sodium acetate-acetic acid buffer (pH 3.5–5.5), Na_2_HPO_4_–NaH_2_PO_4_ buffer (pH 5.5–8.0), Tris–HCl buffer (7.0–8.5), and Gly–NaOH buffer (pH 8.5–10.0), the highest residual activity was taken as 100%; (**b**) the optimum pH of CsnY was determined in the pH range of 3.5 to 10.0 with the buffers above by setting the activity at the optimal pH as 100%; (**c**) the temperature stability of CsnY was assessed by analyzing the residual activity of CsnY after incubation under 0–70 °C for 0.5 and 1 h, with the initial activity as 100%; (**d**) the optimal temperature of CsnY was determined at the range of 0–80 °C, taking the activity at the optimum temperature as 100%.

**Figure 6 marinedrugs-19-00596-f006:**
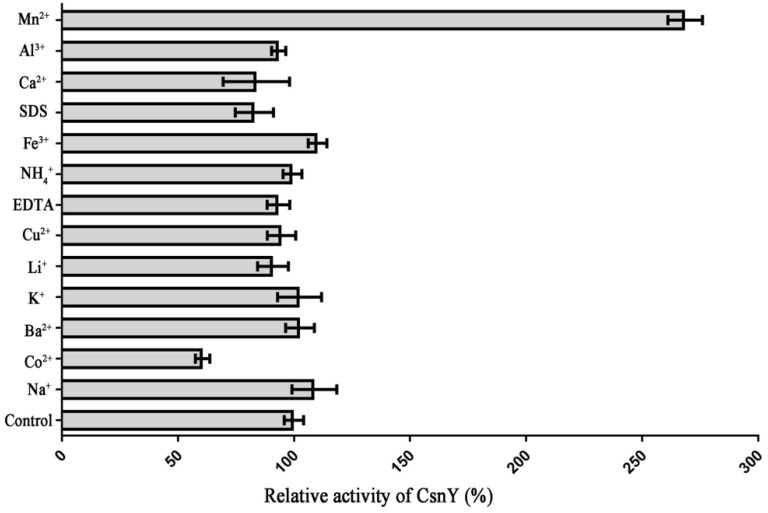
Effects of different metal ions or chemicals on the enzymatic activity of CsnY.

**Figure 7 marinedrugs-19-00596-f007:**
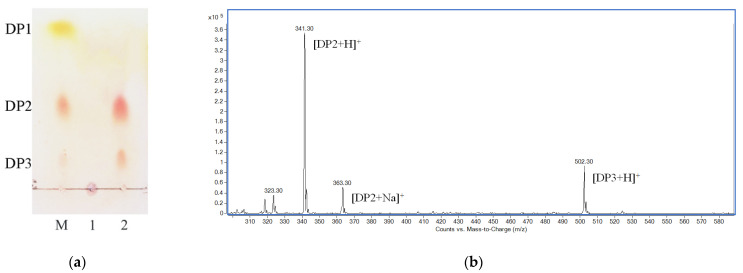
Hydrolysis product analysis by TLC and ESI-MS. (**a**) TLC analysis of end product. Lane M: a mixture of chitomonomer (DP1), -dimer (DP2), and -trimer (DP3); lane 1: mixture before reaction; lane 2: reaction products of CsnY. (**b**) ESI-MS analysis of degraded products.

**Table 1 marinedrugs-19-00596-t001:** Comparison of the properties among CsnY and other cold-adapted chitosanases.

Name/Source	Mw (kDa)	Optimum pH/Temperature (°C)	Specific Activity (U/mg)	Relative Activity at 10/20/30 °C	Thermal Stability	Final Products	Reference
CsnY/*Renibacterium* sp. Y82	27.8	6.0/40	330.67	60%/80%/>80%	80% activity remained after 1 h at 50 °C	DP2,3	This study
CsnB/*Bacillus* sp. BY01	30.89	5.0/35	329.3	40.4%/76.8%/>80%	10% activity retained after 1 h at 40 °C	DP2,3	[15]
CsnM/*Pseudoalteromonas* sp. SY39	28	5.9/40	393.2	30.6%/>50%/80%	only 25.4% and 15.8% activity retained at 30 and 40 °C for 1 h	DP2,3	[16]
Csn-CAP/*Staphylococcus capitis*	35	7.0/30	89.2	N.D./90%/90%	50% activity retained at 55 °C for 1 h	DP2,3	[27]
GsCsn46A/*Gynuella sunshinyii*	29.7	5.5/30	260.39	70%/>80%/>80%	80% activity remained at 30 °C for 1 h	DP2,3	[28]
CsnS/*Serratia* sp. QD07	27.1	5.8/60	412.6	42.6%/40%/>40%	80% activity retained at 30 °C for 2 h	DP2,3	[29]
N.D./*Janthinobacterium* sp. 4239	29	5.0/45	1500	30%/60%/70%	almost all the activity retained after 30 min at 50 °C	DP1–3	[30]

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

*carrageenovora* ASY5. Mar. Drugs.

