# Peer review of "Expression and Characterization of a Novel Cold-Adapted Chitosanase from Marine Renibacterium sp. Suitable for Chitooligosaccharides Preparation"

_marinedrugs, 2021, doi:10.3390/md19110596_

Round 1
Reviewer 1 Report
The authors show the expression of a bacterial chitosanase from the organism Renibacterium sp including very basic bioinformatic analysis and biochemical characterization of this enzyme. These characterizations are, with some exceptions, ok.
However, any novelty, expect that this chitosanase was never characterized before is missing. In my opinion the results shown in this manuscript does not allow to say, that this chitosanase is more suitable to be used in the biotechnical production of defined COS. A comparison to other chitosanases is missing, no long-term stability and no long-term production are shown. The product profile of this enzyme is only very superficially described. It is not clear how much of the starting material or lager COS than D2 and D3 are in the reaction mixture.
Reviewer 2 Report
The article describes the expression of characterization of chitosanase from marine bacterium. It appears to have different properties compared to previously reported enzymes that catalyze the same reaction. However, there are some points that need to be revised, so answer the following questions.
- First, it is stated on line 238 that Renibacterium sp. Y82, which is the origin of the enzyme in this article, was stored in the laboratory, but it seems that there has been no report about the isolation of this strain in academic papers so far. If there is prior reference it should be noted, otherwise the isolation method should be described. Relatedly, it seems that the genome analysis of this strain is also being performed, so this point should be mentioned.
- The pH characteristics of this enzyme and the low temperature adapted enzyme reported so far have been compared (line 203), but the specific activity should be compared.
- Although DNA amplification is usually performed by PCR, the target DNA fragment suitable for codon usage was synthesized in this study. The base length should be written after “the csnY gene” on line 260.
- CDS seems to refer to a mixture of various degrees of polymerization, but in this result, only disaccharide and trisaccharide were produced. From the point of view of preparing CDS, the method using this enzyme seems inappropriate, but please comment on this point.
Minor points
- What’s CDD on line 88?
- Please explain the meaning of black paint, red paint, and blue paint in Fig. 1.
Reviewer 3 Report
This manuscript details results of investigation on a novel cold-adapted chitosanase from the marine Renibacterium and would promote its exploitation in biotechnology for the production of chito-oligosaccharides.
As I read the Special Issue Information paragraph "Tribute to...", where this manuscript has been sent, I wonder about the strict suitability of this manuscript to this collection; however this is left to the SI-editors.
The manuscript contains useful information on a new cold-active marine enzyme and it has been organized in a well manner thus reporting nteresting results. However, as I see it some important flaws are present which are detailed below.
After complete answer to this list the manuscript is acceptable for publication.
line 47 check english why could?
line 56 put lacking reference here at the end of the sentence
line 68-70 please check english here rephrasing the concept in a more clear manner. More words could be used to clarify also the industrial interest for cold-active enzymes with general reference to updated reviews on the topic.
line 144 please briefly report in the result section too, some experimental details: at least concentration of different metal ions or chemicals used.
line 150 As for this experiment more details should be reported, I found the discussion here too much qualitative in nature and no other details seems to be present in Material section (see also below). An interesting addition could be to report results on hydrolysis using different purified chito-oligomers, to better define specificity of the enzyme action.
line 191, 194 put reference here after compared to...
line 196 please be clear adding reference or explaining this vague recalling
line 206-208 here too more words could be helpful to clarify the concepts mentioned
line 210 please add more info about Mn++ effect
line 220-230 the aspect discussed here could be improved with the new experiment suggested above and with quantitative details which are lacking in this manuscript about conversion, time, yields etc.etc.
line 289 since this paragraph contains info about metal salts, this should be clear in the title
line 304 for the whole paragraph more details are necessary not only for the reaction condition, also for reducing sugar analysis, instrumental conditions for ESI-MS etc.etc.
lines 321-322 this final assessments on the excellent properties could be better supported with suggested info and details requested in this review report.
Round 2
Reviewer 1 Report
The second version of the manuscript is, this a few modifications, very similar to the first version. Therefore, I still think that the manuscript has be improved by more data to be published.
The author claim that this enzyme is special, because of its comparable high activity at low temperature, but they did not show the long-term stability of the enzyme, but only over a short time. Additionally, by normalizing the values, it is impossible to see how much of chitosan polymer was already hydrolyzed into the oligomers. The author say how much of the chitosan polymer was hydrolyzed into the oligomers only in the last experiment, but these values (enzyme concentration and substrate concentration) are missing in the pH and temperature experiments. Furthermore, it seems that the authors plotted the activity of the enzyme at 0 °C in the temperature activity and stability experiment. This seems to be very difficult and I'm not sure if this plotting is correct or a slightly higher temperature was used.
In my opinion, it might be a benefit to have only two products, but the separation of them to get one pure form is not easier compared to an enzyme which produce e.g. D2, D3, D4 and D5.
Also, the methods part of the mass spectrometric analysis is missing information and a wrong reference is used her. The authors don't describe how the sample was injected into the system and which MS was used and the English of the new parts has to be improved.
No explanation or hypothesis is given in the discussion why this enzyme has this special low temperature activity or why Mangan leads to an increase of the activity.
Reviewer 2 Report
The points to be revised have been corrected appropriately.
Author Response
We would like to appreciate all the helps and efforts from the editors and reviewers to revise this manuscript. Thanks for all the comments and suggestions.
Reviewer 3 Report
I am happy with all additions and modifications made by authors after the first round of review and with the clear indication of "work in progress" on this enzyme for fine details assessment on the enzyme action within the manuscript itself. Their answers about the additional experiment suggested in my comments on hydrolysis using different purified chito-oligomers are convincing. Indeed they added also "quantitative" details for the experiments reported here and necessary information about experimental section which were both lacking. As such the manuscript reached the quality for publication in my opinion but as indicated before assessment for suitability for the collection in this special issue is left to the SI-editors.
Author Response

(The authors gave the same response as above.)
